# Effects of Shade Net Colors on Mineral Elements and Postharvest Shelf Life and Quality of Fresh Fig (*Ficus carica* L.) under Rain-Fed Condition

**Ali Jokar [1], Hamid Zare [1,2,*], Abdolrasool Zakerin [1] and Abdolhossein Aboutalebi Jahromi [1]**

[1] Department of Horticulture, Jahrom Branch, Islamic Azad University, Jahrom 7414785318, Iran; jokar1342@gmail.com (A.J.); zakerin@jia.ac.ir (A.Z.); aa84607@gmail.com (A.A.J.)

[2] Fig Research Station, Fars Agricultural and Natural Resources Research and Education Center, AREEO, Estahban 7451877802, Iran

\* Correspondence: hameidzare777@gmail.com; Tel.: +98-917-705-1359

**Abstract:** Photoselective netting is well known for filtering the intercepted solar radiation, thus affecting light quality. While its effects on leaf mineral elements have been well investigated, how color netting affects fruit mineral nutrients remains elusive. This study was conducted to evaluate the effects of shade provided by blue and yellow nets on mineral nutrients of fig trees under rain-fed conditions. The experiment was arranged as a split-plot treatment in a randomized complete block design with three replications. Cultivars "Sabz" and "Siah" were covered with color nets or left uncovered (as the control group). The highest nitrogen content (8710 ppm) was recorded for cultivar "Sabz" covered with blue net. Color nets enhanced calcium concentration in cultivar "Siah". Covering fig trees with yellow net increased magnesium content in cultivar "Siah" and phosphorus content in cultivar "Sabz". Our observation showed the significant positive effect of photo selective nets on postharvest quality, by decreasing fig fruit weight loss and extending shelf life of fruits. In general, color nets as a new agro-technological approach can maintain fruit nutrition under rain-fed conditions and increase postharvest shelf life and quality of fresh fig.

**Keywords:** *Ficus carica*; nutrition; shelf life; weight loss





## 1. Introduction

The recent rise in global warming across the world has caused severe challenges to crop production especially in arid and semi-arid areas, including Iran. Among others, the challenges include increase in air temperature, intensity of solar radiation [1], considerable reduction in annual cold nights, increase in the number of annual warm nights [2,3], and scarce water resources [4]. Thus, the extreme environment conditions result in a large increase in transpiration rate, often enhancing root water uptake rate even in well-irrigated soils following reduced photosynthetic activity and adverse effects on plant growth [4]. Therefore, more sophisticated and modern practices are required to alleviate such challenges and dangerous climatic fluctuations in orchards with less energy cost in semi-arid and arid areas. The most effective method would probably be the use of shade nets, because they are able to alter environmental conditions. Photo-selective shade nets can change the light quality by extending the relative proportion of diffuse light (scattered) and also by absorbing different spectral bands [5]. Additionally, the use of different colors of shade net can create desirable microclimates, protect against pests, fungi and physical damage [5], and has the ability to extend the shelf life of fruit, thereby improving the quality of fruit and lowering postharvest losses [6]. The net can also ensure that the air beneath the shade cloth stays humid and restricted, which is of further benefit to the plants by reducing wind damage to the crop and evaporation of soil moisture [7].

"Colored-color nets" (red, yellow, green, and blue) and "neutral-color nets" (pearl, white, and gray) are the two categories of shade net color, absorbing spectral bands either

shorter or longer than the visible range [8]. The blue shade net is designed to absorb the Ultraviolet (UV), Red, and Far red (FR) spectral regions, while enhancing the blue spectral region. Meanwhile, the yellow shade net is designed to substantially decrease the UV and Blue, while elevating the Gray, Yellow, Red, and FR wavelengths [9].

Several studies have reviewed the influence of photo-selective nets on plant vegetative growth, flowering, harvest time (early or late maturation), fruit quality, and yield [8,10]. Netting has developed noticeably in the last decade, and has been applied in different species including blueberry (*Vaccinium corymbosum* L.) [11], apple (*Malus domestica* (Suckow) Borkh.) [12], wine grape (*Vitis vinifera* L.) [13], orange (*Citrus sinensis.* var Valencia) [4], pomegranate (*Punica granatum*) [1], pepper (*Capsicum annum*) [14] and tomato (*Lycopersicon esculentum* Mill.) [8]. Grape vines under shaded conditions exhibit increased vigor, leaf surface and nitrogen content [15]. Shaded apple trees show smaller trunk diameter but increased the number and length of their 1-year shoots under net colors [16]. Additionally, the nets showed some negative effects with respect to the organoleptic characteristics of apples in Italy [17]. The white net decreased soluble solids content (SSC) and flesh firmness and increased starch index in apple fruits at harvest, while reducing the SSC and flesh firmness after cold storage [18]. Photoselective shade nets led to a notable reduced weight loss in tomatoes, red, yellow, and green sweet peppers after postharvest storage [5]. Additionally, fruit firmness was higher in tomatoes and sweet peppers grown under photoselective shade nets after postharvest storage [5,19]. However, a white shade net did not influence weight loss of citrus fruit during storage at −0.6 and 4 °C for 34 days [20].

The fig (*Ficus carica* L.) belongs to the Moraceae family. It is one of the most important horticultural crops cultivated in arid and semi-arid regions of Iran, the largest fig production centers in the world with an annual production of over 87,000 tons [21]. Estahban, located in the south-east of Fars province of Iran, is the biggest dry and fresh fig producer area in Iran. Cultivars "Sabz" and "Siah" are the well-known commercial cultivars of figs in Iran, and are consumed either dry or fresh, respectively. Studies describing the effects of a wide range of photoselective colored nets on the pre- and postharvest quality of cultivars "Sabz" and "Siah" of rain-fed figs are lacking. Therefore, the aim of this study was to investigate the effect of shade net color on nutritional elements and postharvest shelf life and quality attributes of the fresh figs.

## 2. Materials and Methods

### 2.1. Experimental Design and Netting Characteristics

The experiment was conducted at the Fig Research Station, Estahban, Iran (29°08′ N, 54°02′ E, 1760 m altitude). Cultivars "Sabz" and "Siah" (50-year-old plants) were selected. Fig cultivars were planted in a mixture design. Trees were covered with photo-selective shading nets provided by Exsirsaz e Shomal Co. (Babol, Iran). Two colors (blue and yellow) and a control (full sun) were used. Nets were installed horizontally at a height of 3.5 m above each tree on 7 August 2019, and remained on the trees for three months during fruit ripening (i.e., before commercial maturity of the fruits). They were removed after the last fruit harvesting (7 November). The experiment was arranged as a split-plot treatment in a randomized complete block design with three replications. Main plots were two cultivars ("Sabz" and "Siah"), and subplots were the three net colors (blue and yellow net colors and an un-netted control). Each block consisted of two adjacent rows with three trees in each (18 trees in total) at 10 m × 10 m spacing on a sandy loam soil for each cultivar.

### 2.2. Determination of Postharvest Shelf Life and Quality

At harvest time (mid-September), 30 fruits in each replication were harvested at three stages of ripening including, commercially maturity, tree-ripened, and overripe. Figs were considered commercially mature when the fruit flesh was slightly soft when touched [22], and also, at this stage, the skin color of the fruit was light green to yellowish green in the cultivar "Sabz", and reddish-black in the cultivar "Siah". Tree-ripened fruits were riper and softer than commercial maturity, but not overripe. [22]. The flesh of the fruit was

completely softened, and the skin color of "Sabz" fruits had changed to yellow and "Siah" fruit had changed to black. Tree-ripened fruits enter the overripe stage two days later at temperatures between 35 °C and 40 °C. Fig fruits at the overripe stage have grooves in the neck area, and the tissue of this part becomes softer and more wrinkled. Additionally, drying around the fruit ostiole is another feature of this stage. Fruits at this stage usually fall as a semi-dry fruit after three days. Uniform and healthy fruits, free from pests and diseases, were chosen. Then, they were placed in disposable cups without packaging and kept at two different temperatures: in a refrigerator at $4 \pm 1$ °C and at room temperature at $22 \pm 2$ °C for two weeks according to García et al. [23]. The experiment was conducted as a complete randomized design with factorial arrangements [three levels of harvesting stage × two stored temperatures × 6 levels of "Sabz" and "Siah" fig trees were covered with color nets or left uncovered] and three replications. Additionally, orthogonal comparison was used to test the significance of the difference between cultivars and net color.

### 2.3. Observation at Harvest Time

### 2.3.1. Measurements of Shoot Length and Diameter

The total length of the current-year shoot growth and its mean diameter was measured with a ruler and a digital caliper, respectively. In order to measure shoot length and diameter, 10 homogeneous one-year-old shoots distributed around the entire canopy were marked at a height of 1.5 m above the ground.

### 2.3.2. Mineral Analysis

To determine the concentration of potassium ($K^+$), calcium ($Ca^{2+}$), phosphorus (P) and magnesium (Mg) ions in the fruits, oven-dried plant materials were ground to a fine powder. A mass of 0.5 g dry samples was ashed at 500 °C for 8 h and extracted with 5 mL 2 N hydrochloric acid (HCl). After digestion, the volume of the digested samples was made up to 100 mL with distilled deionized water [24]. Concentration of $K^+$ was measured by flame photometry (Jenway PFP7, ELE Instrument Co., Ltd., Stone, Staffordshire, UK) and $Ca^{2+}$ and Mg were measured by atomic absorption spectroscopy (Varian 220, Mulgrave, VIC, Australia). Nitrogen was determined by modified macro-Kjeldahl digestion with the addition of salicylic acid [25].

### 2.3.3. Total Carbohydrates

To determine the concentration of soluble carbohydrates, 150 mg of dried fruit samples was extracted twice with 10 mL ethanol. The samples were centrifuged at 3500 rpm for 10 min, and the volume of the supernatants was adjusted to 25 mL. Concentration of soluble carbohydrate was analyzed according to the method described by Buysee and Merckx [26]. Briefly, 1 mL of the supernatant was transferred to a test tube and 1 mL of phenol 18% was added, along with 5 mL of sulfuric acid (98%). The mixture was shaken immediately, and absorption of the samples was recorded at 490 nm using spectrophotometer (Shimadzu UV 240, Kyoto, Japan). The total sugar concentration of the samples was calculated using calibration curve drawn for glucose standard solutions.

### 2.3.4. Starch Concentration

The starch content of residues left from sugar analysis was determined with Anthron reagent [27]. To this end, 0.2 mL cold distilled water and 0.26 mL perchloric acid 52% (*v/v*) were added to the samples, and the mixtures were shaken for 15 min. After adding 0.4 mL of water, the samples were centrifuged at 5800 rpm for 10 min. The supernatants were separated, and the procedure was repeated with the precipitate. After adding 3 mL of cold Anthron solution 2%, the samples were heated at 100 °C for 20 min. The absorbance of the samples was measured at 620 nm by spectrophotometry (Shimadzu UV 240, Kyoto, Japan).

2.3.5. Fresh Fruit Firmness

Fresh fruit firmness (Kg cm$^{-1}$) was measured on one side of each fruit in commercial maturity stage using an Effegi model FT 327 (Norfolk, VA, USA) manual penetrometer. The results were expressed as mean values of four fruits of each tree.

*2.4. Observation after Storage Period*

2.4.1. Appearance Quality of Fresh Fruit and Shelf Life after Storage

Ten fruits per treatment were used for each quality evaluation. Samples from each treatment were evaluated subjectively after two weeks. Overall quality was evaluated on a 1 to 5 scale according to the overall condition of the fruit, where 1 = unacceptable, 2 = bad, 3 = acceptable, 4 = good, and 5 = excellent. The days when the fruit reached to unacceptable overall quality were recorded and calculated as shelf life. The results were expressed as the shelf life of fresh fruit [28].

2.4.2. Weight Loss

The weight loss of the stored samples was recorded by weighting the samples before and after storage at 4 and 22 °C with digital balance (A&D EJ-303, Seoul, Korea) and the percentage of fruit weight loss was calculated using the following equation:

$$Weight\ loss\ (\%) = \frac{fruit\ weight\ before\ storage\ (g) - fruit\ weight\ after\ storage\ (g)}{fruit\ weight\ before\ storage\ (g)} \times 100$$

*2.5. Determination of Received Energy by the Leaf under the Net*

The wavelengths passing through the yellow and blue nets were considered to be 570 and 490 nm, respectively. The luminous energy (*E*) below each net was calculated using the following formula.

$$E = hf$$

where energy is related with Planck constant [$h = 4.13 \times 10^{-15}$ (eV·s)] and frequency (*f*). Frequency (*f*) is related to wavelength (*λ*) and light speed ($c = 3 \times 10^8$ M·s$^{-1}$) using the following formula.

$$f = c/\lambda$$

*2.6. Statistical Analysis*

Data were analyzed using SAS software (SAS Version 9.4; SAS Institute Inc., Cary, NC, USA.) and significant differences among the mean values were compared by Least Significant Difference (LSD) at *p* < 0.05. Factor analyses were performed for Extraction Method by Principal Component Analysis (PCA). Varimax with Kaiser Normalization was used for the Rotation Method. The relationships between parameters were determined using Pearson's correlation coefficients (r) using SAS software.

**3. Results**

*3.1. Observation at Harvest Time*

3.1.1. Shoot Length and Diameter

Cultivar "Siah" that received full sun showed the highest total length of current-year shoot growth (Table 1). Using the yellow net, "Sabz" cultivar showed the lowest current-year shoot length (5.1 cm) under rain-fed conditions (Table 1).

Colored nets changed the mean of current-year shoot diameter in both cultivars. As shown in Table 1, cultivar "Sabz" covered with blue net possessed the highest mean of current-year shoot diameter (8.1 mm), while uncovered trees reduced the mean of current-year shoot diameter to almost 6.6 and 6.5 mm in cultivars "Sabz" and "Siah", respectively (Table 1).

**Table 1.** The combined effects of color nets and fig cultivars on shoot length and diameter in two fig cultivars.

| Color Nets | Cultivars | Shoot Length (cm) | Shoot Diameter (mm) |
|---|---|---|---|
| Control | "Sabz" | 6.2 [d*] | 6.6 [d] |
| | "Siah" | 8.5 [a] | 6.5 [d] |
| Blue | "Sabz" | 8.2 [ab] | 7.1 [c] |
| | "Siah" | 8.0 [b] | 8.1 [a] |
| Yellow | "Sabz" | 5.1 [e] | 7.0 [c] |
| | "Siah" | 7.3 [c] | 7.4 [b] |

* Different letters in the same column indicate statistically significant differences (significance level $p = 0.05$).

### 3.1.2. Mineral Elements

Covering trees with blue and yellow nets affected nitrogen content in both cultivars. Cultivar "Sabz" showed the highest nitrogen content compared with cultivar "Siah". As shown in Table 2, the highest nitrogen content was recorded for cultivar "Sabz" covered with blue net (8710.0 ppm). Cultivar "Siah" showed the lowest nitrogen concentration under colored nets and uncovered trees.

**Table 2.** The combined effects of color nets and fig cultivars on mineral elements in the dried figs (ppm.).

| Color Nets | Cultivars | Nitrogen | Phosphorus | Potassium | Magnesium | Calcium | N:Ca Ratio |
|---|---|---|---|---|---|---|---|
| Control | "Sabz" | 5836.7 [c*] | 2590 [d] | 10,306 [a] | 15.6 [c] | 3000.0 [d] | 1.97 [b] |
| | "Siah" | 4953.3 [d] | 4200 [c] | 10,025 [a] | 16.0 [bc] | 3200.0 [c] | 1.61 [c] |
| Blue | "Sabz" | 8710.0 [a] | 6900 [b] | 8908 [a] | 17.2 [a–c] | 3333.3 [b] | 2.61 [a] |
| | "Siah" | 4860.0 [d] | 1820 [e] | 8474 [a] | 18.4 [ab] | 3666.6 [a] | 1.34 [d] |
| Yellow | "Sabz" | 6890.0 [b] | 7990 [a] | 9972 [a] | 18.0 [a–c] | 3333.3 [b] | 2.04 [b] |
| | "Siah" | 4903.3 [d] | 1100 [e] | 8901 [a] | 19.2 [a] | 3666.6 [a] | 1.39 [d] |

* Different letters in the same column indicate statistically significant differences (significance level $p = 0.05$).

Phosphorus concentration was significantly affected by the various net color treatments (Table 2). Yellow net increased phosphorus content in cultivar "Sabz", with a value of 7990 ppm, while the lowest content was determined in cultivar "Siah" covered with blue and yellow nets (Table 2).

As shown in Table 2, potassium concentration was significantly affected by color nets in both cultivars. The highest potassium concentration was recorded in cultivar "Sabz" covered with blue nets (12,474.4 ppm). Blue net caused the lowest value of potassium content in cultivar "Siah".

Cultivar "Siah" covered with yellow net had an enhanced level of magnesium content (19.2 ppm). "Sabz" fig trees which received full sun had the lowest magnesium content. Nonetheless, there were no significant differences between the yellow and blue nets in this cultivar and uncovered treatment in cultivar "Siah" (Table 2).

A high concentration of calcium was observed when using blue and yellow nets in cultivar "Siah" (3666.6 ppm) (Table 2). Under full sun, cultivars "Sabz" and "Siah" showed the lowest value of calcium under rain-fed conditions. Altogether, the calcium content in cultivar "Sabz" was lower than in cultivar "Siah" (Table 2).

Nitrogen-to-calcium ratio increased in cultivar "Sabz" under the blue net (2.61), whereas yellow and blue nets in cultivar "Siah" caused the lowest ratio of nitrogen-to-calcium ratio (1.39, 1.34, respectively) (Table 2).

### 3.1.3. Total Carbohydrate and Starch Concentration

Cultivar "Siah" that received full sun ultimately obtained the highest total carbohydrate content (81.7 mg·g$^{-1}$ DW). In comparison, blue nets reduced the total carbohydrate content to almost 69.0 mg·g$^{-1}$ DW in cultivar "Sabz" (Table 3).

**Table 3.** The combined effects of color nets and fig cultivars on total carbohydrate and starch content in the dried figs and fruit firmness in fresh figs.

| Color Nets | Cultivars | Total Carbohydrate (mg·g$^{-1}$ DW) | Starch (mg·g$^{-1}$ DW) | Fresh Fruit Firmness (Kg cm$^{-1}$) |
|---|---|---|---|---|
| Control | "Sabz" | 78.0 [c*] | 5.8 [d] | 10.7 [c] |
| | "Siah" | 81.7 [a] | 11.8 [b] | 11.4 [b] |
| Blue | "Sabz" | 69.0 [e] | 7.6 [c] | 10.7 [c] |
| | "Siah" | 80.0 [b] | 12.3 [ab] | 11.5 [b] |
| Yellow | "Sabz" | 72.7 [d] | 6.8 [c] | 11.6 [b] |
| | "Siah" | 79.7 [b] | 12.8 [a] | 13.4 [a] |

\* Different letters in the same column indicate statistically significant differences (significance level *p* = 0.05).

Colored nets influenced starch content in both cultivars. Cultivar "Siah" being covered with blue and yellow nets had an enhanced starch concentration of 12.8 and 12.3 mg·g$^{-1}$ DW, respectively, whereas uncovered cultivar "Sabz" reduced the starch content to 5.8 mg·g$^{-1}$ DW (Table 3).

### 3.1.4. Fresh Fruit Firmness

Fresh fruit firmness was significantly affected by different color net treatments (Table 3). Yellow nets increased fruit firmness in "Siah" (13.4 Kg cm$^{-1}$), while the softest fruits were recorded in "Sabz" fig trees under blue net and full sun (control) treatments (Table 3).

### 3.1.5. Receiving Luminous Energy by the Leaves

The leaves of the fig trees under the yellow and blue nets received light with a frequency of 530 and 620 terahertz (THz) and energy of 2.17 and 2.6 electron volts (eV), respectively.

### 3.2. Observation after Storage

### 3.2.1. Shelf Life of Fresh Fruit

Fruits of cultivar "Siah" covered with the blue and yellow nets, harvested at commercial maturity and kept at 4 °C, exhibited an extended shelf life under rain-fed conditions. Cultivar "Sabz" receiving full sun and harvested overripe while kept at 22 °C showed the lowest shelf-life. Nonetheless, there was no significant difference between cultivar "Sabz" which received full sun, harvested overripe and kept at 4 °C and "Sabz" fig cultivar covered with the yellow net harvested overripe and kept at 22 °C (Table 4). The orthogonal contrast determined that there were significant differences between "Sabz" vs. "Siah", control vs. yellow and blue nets, control vs. blue net, and control vs. yellow net (Table 5).

### 3.2.2. Fruit Weight Loss

Cultivar "Sabz" covered with the blue net, harvested at commercial maturity stage and kept at 22 °C showed the highest weight loss of 70.1%, while the lowest weight loss was recorded for the "Siah" fig cultivar which received full sun, harvested at the tree ripened stage and kept at 4 °C (20.1%), Meanwhile, there was no significant difference with cultivar "Sabz" covered with the yellow net, harvested at overripe stage and kept at 22 °C (Table 4). In addition, the orthogonal contrast determined that there were no significant differences between the net colors and cultivars (Table 5).

### 3.3. Pearson's Correlation Coefficients between Indices

Pearson's correlation coefficients between shoot length, shoot diameter and sugar, mineral nutrients, firmness, weight loss and shelf life of fruit are presented in Table 6. There was a strong positive correlation (*p* < 0.01) between Ca and diameter, P and K, starch and length, starch and Ca, starch and sugar, firmness and Ca, firmness and starch, shelf life and length, shelf life and diameter, shelf life and Ca, shelf life and starch. A positive correlation (*p* < 0.05) was also observed between Mg and Ca, firmness and Mg. However, there were a strong negative correlation (*p* < 0.01) between sugar and K, sugar and P. A

negative correlation ($p < 0.05$) was detected between Ca and K, starch and K, starch and P, firmness and K, weight loss and sugar (Table 6). The highest relationship was determined between fruit shelf life and starch (r = 0.87).

**Table 4.** The combined effects of color nets, harvest stages and stored temperature on fresh figs weight loss and shelf life.

| Treatment | Harvest Stages | Stored Temperature | Weight Loss (%) | Shelf Life (d) |
|---|---|---|---|---|
| 1 [¥] | Maturity | 4 °C | 28.9 [mn]* | 10 [fg] |
| 1 | Maturity | 22 °C | 65.6 [c] | 7 [ij] |
| 1 | Ripe | 4 °C | 28.2 [n–p] | 8 [hi] |
| 1 | Ripe | 22 °C | 58.6 [e] | 6 [jk] |
| 1 | Over ripe | 4 °C | 23.1 [t] | 5 [kl] |
| 1 | Over ripe | 22 °C | 47.1 [j] | 4 [l] |
| 2 | Maturity | 4 °C | 30.1 [l] | 14 [bc] |
| 2 | Maturity | 22 °C | 70.1 [a] | 11 [f] |
| 2 | Ripe | 4 °C | 28.4 [no] | 12 [de] |
| 2 | Ripe | 22 °C | 65.8 [c] | 10 [fg] |
| 2 | Over ripe | 4 °C | 24.4 [s] | 9 [gh] |
| 2 | Over ripe | 22 °C | 42.9 [k] | 8 [hi] |
| 3 | Maturity | 4 °C | 26.3 [q] | 11 [ef] |
| 3 | Maturity | 22 °C | 68.5 [b] | 8 [hi] |
| 3 | Ripe | 4 °C | 27.6 [op] | 9 [gh] |
| 3 | Ripe | 22 °C | 55.8 [f] | 7 [j] |
| 3 | Over ripe | 4 °C | 20.8 [u] | 6 [jk] |
| 3 | Over ripe | 22 °C | 43.5 [k] | 5 [kl] |
| 4 | Maturity | 4 °C | 23.2 [t] | 14 [bc] |
| 4 | Maturity | 22 °C | 52.2 [g] | 11 [ef] |
| 4 | Ripe | 4 °C | 20.1 [u] | 12 [de] |
| 4 | Ripe | 22 °C | 59.2 [e] | 10 [fg] |
| 4 | Over ripe | 4 °C | 27.4 [p] | 9 [gh] |
| 4 | Over ripe | 22 °C | 51.6 [g] | 8 [hi] |
| 5 | Maturity | 4 °C | 25.3 [r] | 16 [a] |
| 5 | Maturity | 22 °C | 48.6 [i] | 13 [cd] |
| 5 | Ripe | 4 °C | 26.4 [q] | 14 [bc] |
| 5 | Ripe | 22 °C | 65.9 [d] | 12 [de] |
| 5 | Over ripe | 4 °C | 23.3 [t] | 11 [ef] |
| 5 | Over ripe | 22 °C | 55.1 [f] | 10 [fg] |
| 6 | Maturity | 4 °C | 29.4 [lm] | 15 [ab] |
| 6 | Maturity | 22 °C | 65.1 [c] | 12 [de] |
| 6 | Ripe | 4 °C | 28.9 [mn] | 13 [cd] |
| 6 | Ripe | 22 °C | 65.7 [c] | 11 [ef] |
| 6 | Over ripe | 4 °C | 27.9 [op] | 10 [fg] |
| 6 | Over ripe | 22 °C | 50.5 [h] | 9 [gh] |

* Different letters in the same column indicate statistically significant differences (significance level p = 0.05); [¥] 1 = "Sabz" cultivar that received full sun (control group); 2 = "Sabz" cultivar covered with blue nets; 3 = "Sabz" cultivar covered with yellow nets; 4 = "Siah" cultivar that received full sun (control group); 5 = "Siah" cultivar covered with blue nets; 6 = "Siah" cultivar covered with yellow nets.

**Table 5.** Orthogonal contrasts between cultivars and net color in fig trees.

| Treatments | Weight Loss | Shelf Life |
|---|---|---|
| "Sabz" vs. "Siah" | NS | ** |
| Control vs. yellow & blue nets | NS | ** |
| Control vs. blue net | NS | ** |
| Control vs. yellow net | NS | ** |
| Blue net vs. yellow net | NS | NS |

NS, *, ** Nonsignificant and significant at $p < 0.05$ or 0.01, respectively.

**Table 6.** Pearson's correlation coefficients between fig indices under different color nets.

| | Length | Diameter | K | Ca | Mg | P | Sugar | Starch | Firmness | Wight Loss | Shelf Life |
|---|---|---|---|---|---|---|---|---|---|---|---|
| Length | 1 | | | | | | | | | | |
| Diameter | 0.20 | 1 | | | | | | | | | |
| K | 0.09 | −0.43 | 1 | | | | | | | | |
| Ca | 0.27 | 0.85 ** | −0.48 * | 1 | | | | | | | |
| Mg | 0.03 | 0.38 | −0.17 | 0.47 * | 1 | | | | | | |
| P | −0.31 | −0.30 | 0.62 ** | −0.33 | −0.15 | 1 | | | | | |
| Sugar | 0.27 | 0.04 | −0.70 ** | 0.15 | 0.22 | −0.78 ** | 1 | | | | |
| Starch | 0.63 ** | 0.45 | −0.54 * | 0.71 ** | 0.39 | −0.58 * | 0.66 ** | 1 | | | |
| Firmness | −0.05 | 0.29 | −0.52 * | 0.65 ** | 0.51 * | −0.41 | 0.38 | 0.62 ** | 1 | | |
| Weight loss | −0.18 | 0.04 | 0.38 | 0.03 | −0.04 | −0.03 | −0.52 * | −0.38 | 0.09 | 1 | |
| Shelf life | 0.77 ** | 0.65 ** | −0.29 | 0.79 ** | 0.29 | −0.39 | 0.29 | 0.87 ** | 0.40 | −0.16 | 1 |

\* $p < 0.05$ (2-tailed). ** $p < 0.01$ (2-tailed).

### 3.4. PCA and Factor Analysis

Principal components analysis (PCA) was performed to visualize the relation of the measured factors with principal components. Eigenvalues greater than 1 were selected to extract the main principal components. About 46% and 78% of the changes could be justified considering the first component and with three steps, respectively. Starch, shelf life, calcium, sugar and shoot length achieved high values in three components that could be considered as the most effective variables. Three factors had specific values of more than one with factor analysis by Varimax rotation. About 30% and 78% of the changes could be explained by considering the first and three factors in three components, respectively. Calcium, firmness, total sugar, potassium and shelf life of fruit and shoot length and diameter had high coefficients (Figure 1).

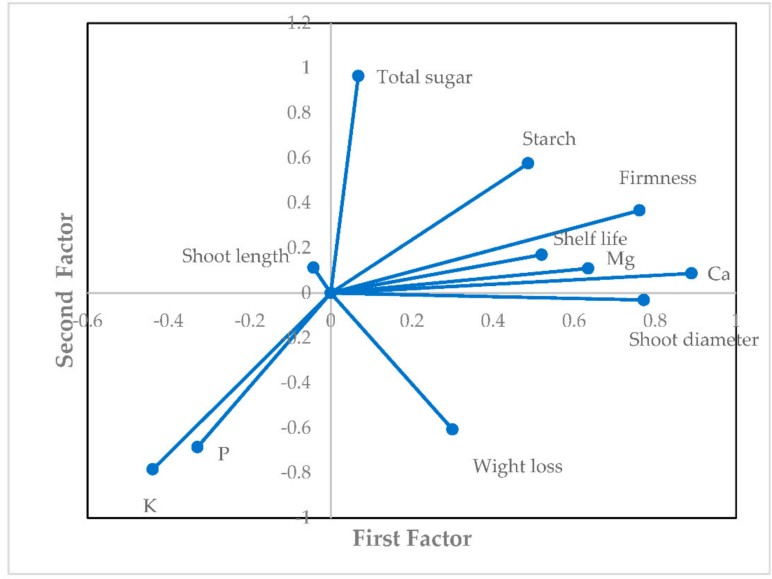

**Figure 1.** Loading plot for factor analysis.

### 4. Discussion

The production and delivery of high-quality fruit in the market remains the key factor in ensuring competitiveness in the fruit industry, thus driving producers to integrate new technology such as photo-selective nets to realize these targets. The influence of shade netting on fruit development and its impact on the postharvest potential of rain-fed fig fruits are chiefly unknown.

Blue shade nets had a positive effect on vegetative growth, manifesting in a significant increase in the current-year shoot growth and shoot diameter over the course of the study. Bastías and Corelli-Grappadelli [29] reported that the shoot growth rate under blue nets was the highest among the different colored nets, which is in accordance with the current study. The longest current-year shoots and smallest shoot diameter under the

blue net in cultivar "Sabz" could be explained by shade avoidance and the photoreceptor, phytochrome, located in meristematic tissue in the shoot tip. In the shade, plants redirect photo assimilates to shoot elongation and away from structures dedicated to resource acquisition and storage, at the expense of leaf development [30]. In line with our study, an increase in mean number of year-old shoots per tree, mean diameter of one-year-old shoots, and total length of year-old shoots in 'Pinova' under 12% white, 14% red-white, 18% red-black and 23% green-black protective netting compared to an uncovered tree were reported by Solomakhin and Blanke [16]. Changes in total solar radiation reaching the tree canopy and changes in leaf gas-exchange under color netting influenced vegetative growth such as shoot diameter. Thus, as blue nets exhibited a substantially smaller R/FR ratio compared to those of yellow color, it seems that this could enhance the mean diameter of the shoots in fig and other fruits such as apple and peach under blue nets [31]. In a related study, it was reported that in warm regions, the decrease in midday leaf temperature and air vapor pressure shortage caused by shading nets increased leaf carbon assimilation, which resulted in an increase in vegetative growth [32].

Though the nutritional characteristics of fig fruit have tremendous benefits in the human diet, the impact of shade netting on these compounds has received little attention.

Shade level affected rain-fed fig fruit mineral nutrient concentration, but the effect differed among nutrients. Fruit N, P, Ca, Mg concentrations increased with shade net, whereas that of K was unaffected (Table 2). There are few studies on the effect of shading on fruit mineral nutrition compared to leaf mineral elements. For instance, a linear decline in water, N and K uptake, and an increase in foliar concentration of N, P, and K with increasing shade level was found in greenhouse tomato; however, fruit mineral nutrient content was not tested [14,33]. Shading increased augmented mineral nutrient concentrations in bell pepper [14].

Leaf nutrient concentrations exhibited variations in species such as apple [34], cherry [35], and kiwifruit [36]. These variations in concentrations could be associated with metabolic and structural functions [37]. Since N, P, Ca and Mg are the mobile elements, they can translocate to fruit immediately, resulting in an increase in fruit size [38]. Additionally, N, P, K, Ca, and Mg mineral concentrations increased by 20% on average in shaded plants; the increased mineral nutrients were closely associated with decreased carbohydrate accumulation [15]. In line with the current results, the higher $Ca^{2+}$ concentration observed in shaded apple fruit tissue was thought to be attributed to lower dilution of the $Ca^{2+}$ accumulated in the fruit due to lower fruit growth rates [39]. Therefore, other factors are also responsible for the higher $Ca^{2+}$ concentration observed in the cortical tissue of shaded fruit [39]. Our results indicate that the highest phosphorus concentration belongs to cultivar "Sabz" covered with yellow net, so it seems that changes in phosphorus concentration can be cultivar dependent.

Starch accumulation in fruits under color shade nets resulted in less sweet fruit than those receiving full sun (Table 3). Results from other studies also indicate that the lower sugar content of fruit produced in the colored shade nets may be associated with lower light intensity; nevertheless, the photosynthetic system should not be stopped. Moreover, greater light intensity is able to enhance the photosynthetic activity of the plants [1,40]. According to the findings of Solomakhin and Blanke (2010), the extreme tree vigor caused by the colored nets is related to reduction of red to far red (R: FR) ratio, which regulates the activity of phytochrome in leaf and fruit. Therefore, a decrease in invertase activity under low light intensity occurs [40].

Photo-selective nets had significant effects on fruit firmness in rain-fed fig fruits (Table 3), which was in agreement with previous research on tomato fruit [41]. It has been reported that growing tomato plants under color nets and in optimal conditions can result in firmer fruits with a thicker pericarp and a better tolerance to transport than control [41]. According to Campbell et al. [42], a softer flesh was observed in apples under shading conditions. They suggested that a decrease in flesh firmness might be the result of poor activity in cell wall formation and a high water influx to fruit cortex cells.

The present study showed that photo-selective nets along with ripening stages decreased fruit weight loss in both fig cultivars resulting in extending shelf life. Weight loss due to water decrease results in a decrease of freshness and fruit firmness, and affects the shelf life and consumer acceptance [5,43]. It seems that when fruits are harvested at commercial maturity, the flesh firmness is the highest compared with the other ripening stages, resulting in an increase in the fruit shelf life (Table 4).

In the current study, fig fruits harvested at the tree ripened and overripe stages showed the lowest weight loss. This lower percentage of weight loss could be ascribed to a lower degree of moisture loss due to the fruit ripening development known to be a characteristic of fruit ripening and customer acceptance. Additionally, during the postharvest period, weight loss mostly depends on the storage temperatures [44], and our study confirmed that the weight loss was mainly affected by storage temperatures, whereby the fruits stored at 4 °C showed less weight loss than those stored at room temperature (Table 4). Higher storage temperature may increase more moisture loss which is in line with the previous report on mandarin fruit [43].

The relationship between Ca and fruit firmness in this study is thought to be related to an important function of Ca in plants in increasing the rigidity of the cell wall and promoting cohesion of neighboring cells. Magnesium can also play an important role in fruit firmness similar to calcium. However, this statement contradicts the findings of Hopkirk et al. [45]. It seems that an increase in fruit firmness could be associated with high concentrations of starch, resulting in extended fig fruit shelf life. According to our results, the negative correlation between phosphorus and starch and sugar content may be due to its role in sugar metabolism. If inorganic phosphorus concentration is too high, it can show an antagonism effect, and $CO_2$ fixation can also be negatively affected [37,46].

Our results demonstrate that the effects of net colors are variable dependent, and are different among the studied parameters. Based on the factor analysis, calcium, firmness, total sugar, potassium and shelf life of fruit, as well as shoot length and diameter, showed the best response under net coverings and were considered the most effective variables.

## 5. Conclusions

The results of our study showed that two net colors had different influences on the fruit minerals of two cultivars. Hence, the genetic backgrounds of different cultivars play an important role in fruit minerals uptake. There was an increase in the N and K fruit concentrations and nitrogen-to-calcium ratio in cultivar "Sabz" covered with blue net. Yellow net enhanced phosphorus content in cultivar "Sabz" and magnesium content in cultivar "Siah". A high concentration of calcium was recorded when using blue and yellow nets in cultivar "Siah". Fig trees receiving full sun possessed enhanced total carbohydrate content and reduced starch content. Fruit weight loss rate decreased and fruit shelf-life was prolonged in fig cultivars harvested at commercial maturity during storage. Based on the factor analysis, calcium, firmness, total sugar, potassium and shelf life of fruit, as well as shoot length and diameter, showed the best response under net coverings and were considered to be the most effective variables. In general, color nets, as a new agro-technological approach, can maintain fruit nutrition under rain-fed conditions and enhance appearance quality of fresh figs.

**Author Contributions:** Conceptualization, A.J., H.Z., and A.Z.; Project administration, H.Z., A.Z., and A.A.J.; Supervision, H.Z., and A.Z.; methodology, A.J., and H.Z.; validation, A.J., H.Z., A.Z., and A.A.J.; Software, A.J., and H.Z.; formal analysis, A.J., and H.Z.; investigation, A.J., and H.Z.; resources, A.J., and H.Z.; data curation, A.J., and H.Z.; writing—original draft preparation, A.J.; writing—review and editing, H.Z., A.Z., and A.A.J.; visualization, A.J., and H.Z. All authors have read and agreed to the published version of the manuscript.

**Funding:** This research received no external funding.

**Institutional Review Board Statement:** Not applicable.

**Informed Consent Statement:** Not applicable.

**Data Availability Statement:** Data is contained within the article.

**Acknowledgments:** The authors wish to extend their thanks and appreciation to Estahban Fig Research Station.

**Conflicts of Interest:** The authors declare no conflict of interest.

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
