# Peer review of "Effects of Shade Net Colors on Mineral Elements and Postharvest Shelf Life and Quality of Fresh Fig (Ficus carica L.) under Rain-Fed Condition"

_horticulturae, doi:10.3390/horticulturae7050093_

Round 1

Reviewer 1 Report

This study is aimed to investigate the effect of shade net color on nutritional elements and postharvest shelf life and quality attributes of the fresh figs. The study is designed in an acceptable level. The study can provide some new elements. There are some points to correct and data (correlations) need to be further analysed. The study can be suitable for publication after sufficient revisions. English should be corrected throughout the ms.

Specific suggestions and comments

L23: Cultivars “Sabz” and “Siah”. L74 és L77. Check throughout the whole text.

L24: cultivar “Sabz” and not “Sabz” cultivar. The same L26-28. Check throughout the whole text.

L81: Delete points after the titles. Throughout the whole text.

L87: 7 August, 2019

L89: 7 November

L145: ’2.4.1. Appearance quality of fresh fruit after shelf life’ and not ’2.4.1. Shelf life of fresh fruit.’

L148: Give short description of the appearance quality

L157: P < 0.05.

L173: Table 1: space is missing between ’diameter’ and ’(mm)’

L194: Table 2: ’Phosphor’ - this is not English

L226: I suggest to prepare a correlation between parameters not only by Pearson correlation but also by factor analyses.

L275. space between ’40]’ and ’. According…’

L304: Give the conclusions in points.

L320: References do not follow consistent format e.g. Journal names italic or not intalic?

L381: issue and page number are missing

Author Response

Dear Reviewer 1

Thank you for giving us the opportunity to revise the manuscript. I am also grateful for the constructive comments on the manuscript.

Please find responses as follows with red color:

Moderate English changes required.  English language was edited according to reviewer comments throughout the manuscript.

Is the research design appropriate? Can be improved. The research design was corrected in lines 89-91.

Are the methods adequately described? Can be improved. Materials and Method section was strengthened (lines 116- 117, 146- 151, 157-164, and 169-170).

Are the results clearly presented? Can be improved. Results section was improved (lines 211- 213, and pages 8-11).

Are the conclusions supported by the results? Must be improved. The conclusions were improved (lines 333- 338).

English should be corrected throughout the ms. English language was moderately edited throughout the manuscript.

Specific suggestions and comments

L23: Cultivars “Sabz” and “Siah”. L74 és L77. Check throughout the whole text. The whole text was check and edited.

L24: cultivar “Sabz” and not “Sabz” cultivar. The same L26-28. Check throughout the whole text. The whole text was check and edited.

L81: Delete points after the titles. Throughout the whole text. The points after the titles were deleted.

L87: 7 August, 2019. Date report was corrected in line 87.

L89: 7 November. Date report was corrected in lines 88- 89.

L145: ’2.4.1. Appearance quality of fresh fruit after shelf life’ and not ’2.4.1. Shelf life of fresh fruit.’ The title was corrected in line 146.

L148: Give short description of the appearance quality. The description of the appearance quality was explained in lines 147-151.

L157: P < 0.05. It was corrected in line 169.

L173: Table 1: space is missing between ’diameter’ and ’(mm)’. Space was created between ’diameter’ and ’(mm)’ in Table 1.

L194: Table 2: ’Phosphor’ - this is not English. It was edited in Table 2.

L226: I suggest to prepare a correlation between parameters not only by Pearson correlation but also by factor analyses. Factor analyses were added in lines 169- 170, and pages 8-11.

L275. space between ’40]’ and ’. According…’ It was edited in line 302.

L304: Give the conclusions in points. The conclusions were improved (lines 333- 338).

L320: References do not follow consistent format e.g. Journal names italic or not intalic? References were corrected in pages 14-17.

L381: issue and page number are missing. They were added in line 412.

Reviewer 2 Report

Methodology: light wave length and energy can be measured with the change of the net, which would make more sound-science.

Discussion: At least authors shall discuss relevantly on those elements of light conditions.

References: Reference section should be carefully edited.

Author Response

Dear Reviewer 2

Thank you for giving us the opportunity to revise the manuscript. I am also grateful for the constructive comments on the manuscript.

Please find responses  as follows with red color:

Moderate English changes required.  English language was edited according to reviewer's comments throughout the manuscript.

Is the research design appropriate? Can be improved. The research design was corrected in lines 89-91.

Are the methods adequately described? Can be improved. Materials and Method section was strengthened (lines 116- 117, 146- 151, 157-164, and 169-170).

Are the results clearly presented? Can be improved. Results section was improved (lines 211- 213, and pages 8-11).

Are the conclusions supported by the results? Can be improved. The conclusions were improved (lines 333- 338).

Methodology: light wave length and energy can be measured with the change of the net, which would make more sound-science. Received wavelengths and energy by the leaf under the nets were added in lines 157- 165 and 211-213.

Discussion: At least authors shall discuss relevantly on those elements of light conditions. The most important elements (calcium, and potassium) were determined based on the PCA analysis. Discussion was strengthened in lines 327- 329.

References: Reference section should be carefully edited. References were carefully edited in pages 14-17.

Round 2

Reviewer 1 Report

Authors did sufficient revisions.

Author Response

Dear Reviewer of “Horticulturae” Journal

We are grateful for your constructive comments on the manuscript.

Please find responses to your comments as follows and the manuscript changes are displayed in red.

The comments and their responses

  1. The authors mentioned four traits, one from each rotated component, however, in some components, there are more than one traits have a slightly lower loading value (but the value is still high, like >0.8) compare to that of the author mentioned trait in the same component. We mentioned attributes with a value higher than 0.7 and the diagram is only related to the first and second components and the first and second factors, so some of the factors or attributes that we have written, they may effectively be small in this diagram, because it shows only the effective elements in the first two components or factors.
  2. It is unclear why the trait loading plot values for each trait in Fig 1 and Fig2 are different than the loading value listed in the Table 7 and Table 8, respectively. The authors need clarification. The answer is similar to response of the reviewer's first comment.
  3. In Table 6, the K is not significantly associated with any other traits, statistically, this trait should be removed before the PCA analysis. A potassium data point is changed (the non-significant effect in the previous version was due to the mistake in entering a data point.) and its relationships is corrected in Table 6 of the manuscript.
  4. If the authors want to present something from this analysis, they may delete Table 8 and 7, plus Fig 1, just keep Fig 2. Tables 7 and 8 and Figure 1 were deleted, only Figure 2 is kept and corrected.
  5. There may be typos in the TABLE 8, some FA1 should be FA2, FA3 and FA4. Tables 8 was deleted.

Minor issues:

Table 7 title: "matrixa" should be matrices"? Tables 7 was deleted.

Table 7 and 8: The first character of mineral names should be capitalized. Tables 7 and 8 and Figure 1 were deleted, only Figure 2 is kept and corrected.

There are many grammar errors. For example, line 96: "gave slightly"; line 146: after shelf-life"; line 333:"there was an increased"; line 336: shelf-life lengthened... Suggest to be edited by an English speaker. The entire text was reviewed and edited.

The best regards
